# Self-prompted Chain-of-Thought on Large Language Models for Open-domain Multi-hop Reasoning

**Jinyuan Wang**[1, 3] and **Junlong Li**[2, 3] and **Hai Zhao**[2, 3*]

[1]SJTU-Paris Elite Institute of Technology, Shanghai Jiao Tong University
[2]Department of Computer Science and Engineering, Shanghai Jiao Tong University
[3]Key Laboratory of Shanghai Education Commission for Intelligent Interaction
and Cognitive Engineering, Shanghai Jiao Tong University
{steve_wang,lockonn}@sjtu.edu.cn, zhaohai@cs.sjtu.edu.cn

## Abstract

In open-domain question-answering (ODQA), most existing questions require single-hop reasoning on commonsense. To further extend this task, we officially introduce open-domain multi-hop reasoning (ODMR) by answering multi-hop questions with explicit reasoning steps in open-domain setting. Recently, large language models (LLMs) have found significant utility in facilitating ODQA without external corpus. Furthermore, chain-of-thought (CoT) prompting boosts the reasoning capability of LLMs to a greater extent with manual or automated paradigms. However, existing automated methods lack of quality assurance, while manual approaches suffer from limited scalability and poor diversity, hindering the capabilities of LLMs. In this paper, we propose Self-prompted Chain-of-Thought (SP-CoT), an automated framework to mass-produce high quality CoTs of LLMs, by LLMs and for LLMs. SP-CoT introduces an automated generation pipeline of high quality ODMR datasets, an adaptive sampler for in-context CoT selection and self-prompted inference via in-context learning. Extensive experiments on four multi-hop question-answering benchmarks show that our proposed SP-CoT not only significantly surpasses the previous SOTA methods on large-scale (175B) LLMs, but also nearly doubles the zero-shot performance of small-scale (13B) LLMs. Further analysis reveals the remarkable capability of SP-CoT to elicit direct and concise intermediate reasoning steps by recalling ∼50% of intermediate answers on MuSiQue-Ans dataset.

## 1 Introduction

Open-domain question-answering (ODQA) is a longstanding and challenging task which addresses factoid commonsense questions without specific contexts provided. While existing works in ODQA primarily focus on resolving questions that mostly require single-hop reasoning, there is a burgeoning interest in multi-hop question-answering (MHQA), which aims to derive the correct answer through multi-step reasoning over a collection of candidate articles (Mavi et al., 2022). Yet, a significant disparity exists between such scenarios and real-world applications, since the latter often lacks an explicit set of candidate articles provided by users. In light of this, we officially introduce open-domain multi-hop reasoning (ODMR) as a progression task of ODQA, which requires MHQA with explicit rationales in open-domain setting.

For ODMR, an emerging approach is to leverage large language models (LLMs) due to the vast knowledge stored within their numerous parameters. In recent years, LLMs have shown powerful reasoning and instruction-following capabilities, such as GPT-3 (Brown et al., 2020), PaLM (Chowdhery et al., 2022) and InstructGPT (Ouyang et al., 2022). After extensive training on vast corpora of textual data, LLMs prove to be zero-shot reasoners on complex reasoning tasks by breaking down multi-step questions into intermediate ones for step-by-step reasoning before producing the final answer (Kojima et al., 2023). Such series of intermediate reasoning steps is known as chain-of-thoughts (CoTs) (Wei et al., 2023). CoTs often serve as in-context demonstrations for in-context learning (ICL) (Brown et al., 2020), which enables LLMs to generate outputs that are formally consistent with a target task via a few reference examples provided as prompt. Manual-CoT (Wei et al., 2023) adopt manually designed CoTs as in-context demonstrations to improve the reasoning performance of LLMs. However, it demands delicate and meticulous design by humans, and the demonstrations are the same for each question, which may be sub-optimal. Zero-shot-CoT (Kojima et al., 2023) was proposed to trigger automated CoTs

---

*Corresponding author. This paper was partially supported by the Joint Research Project of Yangtze River Delta Science and Technology Innovation Community (No. 2022CSJGG1400).

by certain specific prompting techniques, such as `"Let's think step by step:"`. Zhang et al. (2022) proposed Auto-CoT, an automated framework to mass-produce CoTs and build in-context demonstrations. However, previous works have not fully leveraged the strong instruction-following and zero-shot reasoning capabilities of LLMs.

In this paper, we propose Self-prompted Chain-of-Thought (SP-CoT), an LLM-only framework to mass-produce high-quality CoTs for ODMR. In general, SP-CoT introduces an automated generation pipeline of ODMR datasets, an adaptive sampler for CoT selection and self-prompted inference via ICL. The automated ODMR datasets are MHQA datasets without candidate contexts, yet including multi-hop questions with six types of complex reasoning chains and step-by-step decomposition. Each intermediate QA step is equipped with a short explanation to justify the answer. By leveraging the ICL ability of LLMs, our method is generally effective on LLMs of different scales.

We evaluate our method on four MHQA datasets in an open-domain setting: ComplexWebQuestions (CWebQ) (Talmor and Berant, 2018), HotpotQA (Yang et al., 2018), 2WikiMultiHopQA (2Wiki) (Ho et al., 2020) and MuSiQue-Ans (MSQ) (Trivedi et al., 2022). Extensive experiments show that our proposed SP-CoT not only significantly surpasses the previous SOTA methods on large-scale (175B) LLMs, but also nearly doubles the zero-shot performance on small-scale (13B) LLMs in ODMR. Further analysis reveals the outstanding capability of SP-CoT to elicit direct and concise intermediate reasoning steps by recalling ∼50% of intermediate answers on MSQ dataset.

Our contributions can be summarized as follows:

1. We introduce an automated pipeline to generate high-quality ODMR datasets by LLMs, which include 2-4 hop questions with six types of complex reasoning chains.

2. We propose SP-CoT, an automated framework to mass-produce CoTs while ensuring quality and diversity.

3. We conduct extensive experiments to confirm the effectiveness of SP-CoT on four ODMR benchmarks. In ODMR setting, our approach significantly boosts the performance by eliciting high-quality intermediate reasoning steps.

Our code and datasets are publicly available at https://github.com/noewangjy/SP-CoT.

## 2 Related Works

### 2.1 Multi-Hop Dataset Creation

Creating an annotated MHQA dataset manually requires significant human resources. Therefore, some researchers are dedicated to automating the generation of MHQA datasets. Jiang et al. (2020) elaborated the creation of a multi-hop fact verification dataset from existing HotpotQA dataset. Trivedi et al. (2022) introduced a bottom-up process to build challenging multi-hop reading comprehension QA dataset through meticulous selection and composition of single-hop questions derived from existing datasets. Press et al. (2023) proposed an automatically generated dataset with compositional 2-hop questions about celebrities. Nevertheless, existing approaches are either only partially automated, still requiring crowdsourcing, or they are limited to less complex 1-2 hop questions. In this work, our proposed SP-CoT is capable of automatically generating 2-4 hop questions with six different types of reasoning chains (Figure 6 in Appendix).

### 2.2 Chain-of-Thought Prompting

Recent works on CoT prompting can be divided into two research lines. The first is prompting LLMs step by step to leverage their comprehension and reasoning abilities to answer questions. Zero-shot-CoT (Kojima et al., 2023) adopts a two-stage design, which requires LLMs to first generate intermediate rationale and then produce an answer. Wang et al. (2022) introduced iCAP, which iteratively prompts a fine-tuned small-scale LLM to generate CoTs and then combines the generated rationales to formulate answers. Least-to-Most (Zhou et al., 2023) requires LLMs to first decompose a complex question into sub-questions and then sequentially solve them to arrive at the final answer.

The second research line focuses on designing effective CoT as demonstrations for ICL to release more powerful reasoning abilities of LLMs. Manual-CoT (Wei et al., 2023) was introduced to leverage manually designed CoT as in-context demonstrations to solve arithmetic, commonsense, and symbolic problems. A recent work (Zelikman et al., 2022) shed light on the practicality to automate the generation of rationales by LLMs. Subsequently, Self-Ask (Press et al., 2023) was proposed to construct step-by-step demonstrations with explicit decision process and intermediate answers

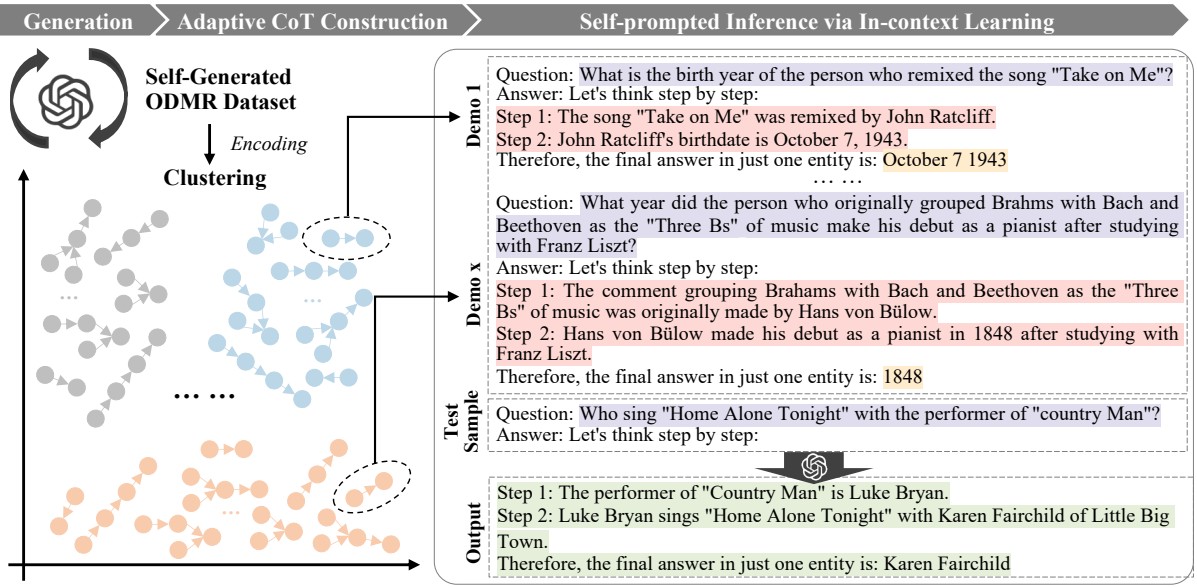



**Generation** > **Adaptive CoT Construction** > **Self-prompted Inference via In-context Learning**

Self-Generated ODMR Dataset

*Encoding*

**Clustering**

**Demo 1**
Question: What is the birth year of the person who remixed the song "Take on Me"?
Answer: Let's think step by step:
Step 1: The song "Take on Me" was remixed by John Ratcliff.
Step 2: John Ratcliff's birthdate is October 7, 1943.
Therefore, the final answer in just one entity is: October 7 1943
… …

**Demo x**
Question: What year did the person who originally grouped Brahms with Bach and Beethoven as the "Three Bs" of music make his debut as a pianist after studying with Franz Liszt?
Answer: Let's think step by step:
Step 1: The comment grouping Brahams with Bach and Beethoven as the "Three Bs" of music was originally made by Hans von Bülow.
Step 2: Hans von Bülow made his debut as a pianist in 1848 after studying with Franz Liszt.
Therefore, the final answer in just one entity is: 1848

**Test Sample**
Question: Who sing "Home Alone Tonight" with the performer of "country Man"?
Answer: Let's think step by step:

**Output**
Step 1: The performer of "Country Man" is Luke Bryan.
Step 2: Luke Bryan sings "Home Alone Tonight" with Karen Fairchild of Little Big Town.
Therefore, the final answer in just one entity is: Karen Fairchild


Figure 1: The overall framework of our proposed SP-CoT, including an automated generation of ODMR datasets, an adaptive sampler for CoT selection and self-prompted inference via ICL. Texts highlighted in purple refer to questions, in red to previously generated CoTs, in orange to answers, and in green to newly generated contents.

as CoT. Zhang et al. (2022) proposed Auto-CoT, which automatically constructs CoTs via LLMs and adopts clustering methods to dynamically build demonstrations for each question.

However, existing methods have two significant limitations: 1) Over-reliance on the reasoning abilities of LLMs. Most methods are reported effective on large-scale LLMs like InstructGPT, while reproducing these methods on small-scale LLMs is quite challenging. 2) Over-confidence on the quality of intermediate results. When prompting LLMs step by step, defects in previous steps may limit the performance of subsequent steps. Similarly, while automatically constructing in-context demonstrations, the effectiveness of ICL might be compromised by the unstable quality of CoTs. Admittedly, manually constructed CoTs can ensure quality, yet they face a trade-off between content diversity and costs. To overcome the above drawbacks, our proposed SP-CoT automates CoT generation with quality ensured by leveraging the strong instruction-following capability of LLMs.

### 2.3 Model Enhancement via LLM Generation

With the powerful capabilities of LLMs on content generation and instruction-following, one recent research direction extensively leverages the content generated by LLMs to enhance smaller LLMs. Recent works such as GPTeacher,[1] Al-

paca (Taori et al., 2023) and Vicuna (Chiang et al., 2023) collect the content generated by GPT-4 (OpenAI, 2023) and the corresponding prompts to train smaller-scale LLMs to achieve comparable performance. Another research line aims to boost the performance of large-scale LLMs to higher levels by leveraging the self-generated content. Some works use the self-generation as contexts to assist themselves in answering questions, such as eliciting intermediate rationales as CoT (Kojima et al., 2023) or generating background articles for reading comprehension (Yu et al., 2023). While others instruct LLMs to generate demonstrations for ICL during inference (Zhang et al., 2022), such as prompting LLMs to generate reliable QA pairs as self-prompted in-context demonstrations (Li et al., 2022). Our work is dedicated to extending the second research line to ODMR by leveraging the automated self-generated CoT as in-context demonstrations. Compared to previous works (Kojima et al., 2023; Zhang et al., 2022; Li et al., 2022), our work taps into the potential of self-prompting LLMs with more complicated framework design to solve a most challenging task.

## 3 Methods

In this section, we elaborate our proposed SP-CoT in three stages (Figure 1):

In the first stage, we prompt LLMs to iteratively generate 2-hop commonsense QA quadruplets with

---

[1] https://github.com/teknium1/GPTeacher

**Stage 1: 2-Hop QAs via Self-Generation**

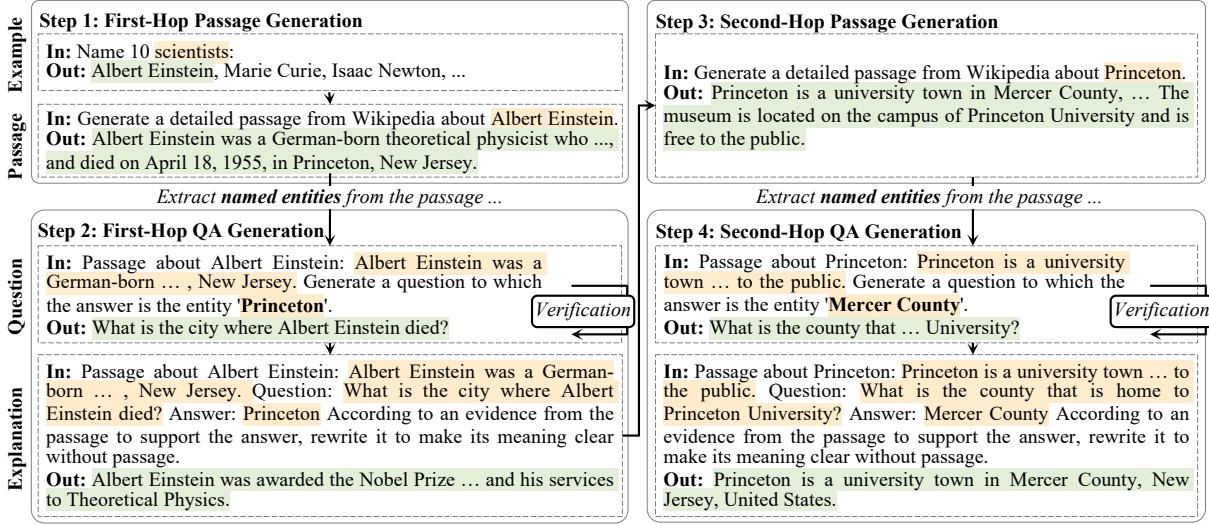

Figure 2: Generation steps for 2-hop QA quadruplets. Each QA quadruplet comprises a question $q$, its corresponding answer $a$, and a context passage $p$. The explanation $e$ includes the answer $a$ from the passage $p$ to address the question $q$. Text highlighted in orange refers to previously generated content, while the response of the LLM is highlighted in green.

context, question, answer and explanation.

In stage 2, we construct multi-hop reasoning chains by connecting 2-hop QA quadruplets and build an ODMR dataset via composition.

In the last stage, we adopt clustering-based sampling approach to dynamically select and construct in-context demonstrations for inference.

### 3.1 2-Hop QAs via Self-Generation

In the first stage, we prompt LLMs to iteratively generate 2-hop QA quadruplets with context, question, answer and explanation, which is illustrated in Figure 2. Inspired by Li et al. (2022), we design a 2-hop commonsense QA generation pipeline, including the following 4 steps:

**Step 1: First-Hop Passage Generation** To guarantee the comprehensive coverage of commonsense knowledge, we manually design 29 diverse topics based on the statistics of TriviaQA (Joshi et al., 2017). For each topic, we require the LLM to name a certain number of keywords. For each collected keyword $k_1$, we ask the LLM to generate a Wiki-style passage $p_1$. Despite some factoid errors (Li et al., 2022), such generated passages contain sufficient factual information to serve as context for QA generation.

**Step 2: First-Hop QA Generation** Given that the answers for commonsense questions are likely

to be named entities, we use Spacy[2] and NLTK (Bird and Loper, 2004) libraries to extract the named entities in the passage $p_1$ as candidate answers. For each candidate answer $a_1$, we require the LLM to raise a question $q_1$ to which the answer is $a_1$ based on the passage $p_1$. To ensure the quality of $q_1$, we employ a double-check process, where we demand the LLM to answer the generated question $q_1$ given the context $p_1$ to check if the generated answer $a_1'$ is accordant with $a_1$. Once the generated QA pair passes the double-check, we prompt the LLM to write a short explanation $e_1$ for it. Note that the candidate answers must exclude the keyword ($a_1 \neq k_1$) because the answer in the first hop will become the keyword for the second hop ($k_2 = a_1, k_2 \neq k_1$). In addition to that, a valid explanation must contain the answer ($a_1 \in e_1$).

**Step 3: Second-Hop Passage Generation** Before the first-hop answers are used as keywords for second-hop passage generation, we use Spacy to filter out the answers with certain labels (QUANTITY, ORDINAL, CARDINAL, PERCENT, MONEY, DATE, TIME), which are infeasible for Wiki-style passage generation. Given a keyword $k_2$, we repeat the same prompts as described in Step 1 to generate the passage $p_2$.

**Step 4: Second-Hop QA Generation** We be-

---

[2]https://spacy.io/

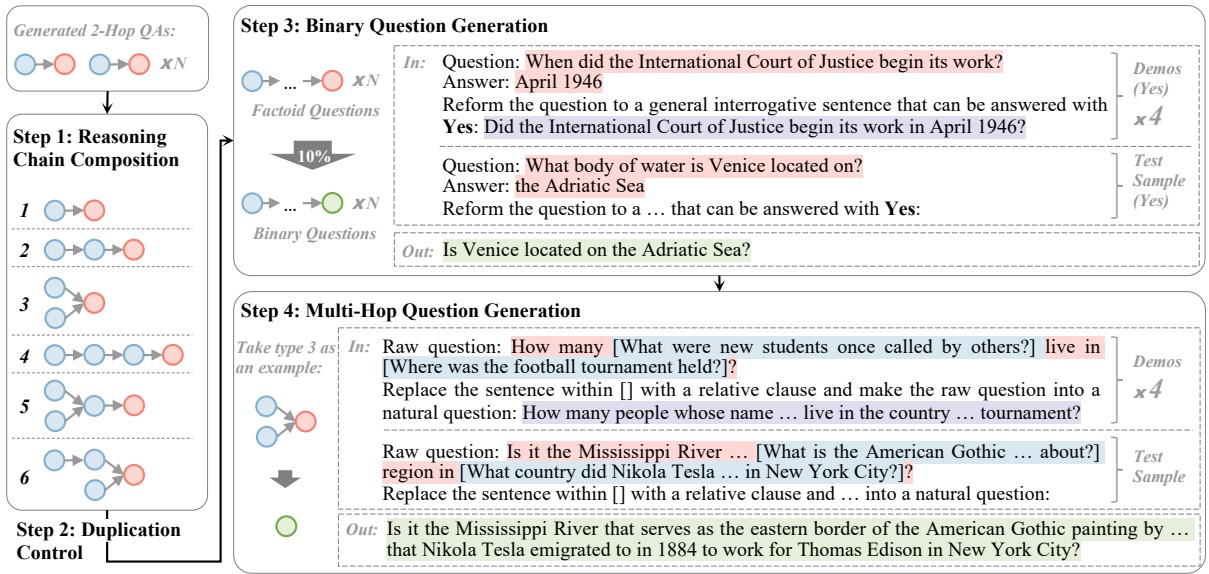

Figure 3: Generation steps for MHQA groups. In step 3 and step 4, we use 4 manually designed demonstrations for ICL. Each MHQA group includes a multi-hop question, the corresponding answer and decomposed QA quadruplets. Nodes and texts highlighted in red, blue and green successively refer to the last hops, intermediate hops and generated hops. Manually designed texts are highlighted in purple.

gin with extracting candidate answers in the generated passage $p_2$ while blocking the keyword $k_1$ and the answer $a_1$ (also known as $k_2$) in the first-hop QA to avoid cyclic graphs. For each candidate answer $a_2$, we require the LLM to generate a question $q_2$ which contains the first-hop answer $a_1$ and can be answered by the candidate answer $a_2$. We examine the quality of $q_2$ with the same double-check in Step 2, and ensure the second-hop question $q_2$ contains the first-hop answer $a_1$ ($a_1 \in q_2$) for connected reasoning. Then we repeat the same prompts in Step 2 to generate explanation $e_2$.

So far, we have instructed the LLM to generate a 2-hop commonsense QA quadruplet pair, which is $(p_1, q_1, a_1, e_1) \rightarrow (p_2, q_2, a_2, e_2)$ with $a_1 \in q_2$. Detailed prompt templates are shown in Figure 2 and Appendix B.

## 3.2 Multi-Hop QAs via Composition

In stage 2, we construct multi-hop reasoning chains with the connected 2-hop QA quadruplets, which is illustrated in Figure 3. We propose an automated dataset construction pipeline to build ODMR datasets with 2-4 hops, which has the following 4 steps:

**Step 1: Reasoning Chain Composition** To connect more questions, we follow the composability criteria (Trivedi et al., 2022), that is, two single-hop QA pairs $(q_1, a_1)$ and $(q_2, a_2)$ are composable into

a multi-hop question $Q$ with $a_2$ as a valid answer if $a_1$ is a named entity and it is mentioned in $q_2$. Such criteria are already satisfied when our 2-hop QA pairs are generated, we use this criterion for connecting more questions. We adopt 6 reasoning graphs with 2-4 hops to build 6 types of multi-hop reasoning chains (Figure 6 in Appendix), and we ensure that in each reasoning chain: 1) the answer $a_i$ to an intermediate question $q_i$ will appear and ONLY appear in its next-hop question $q_{i+1}$ to avoid shortcuts; 2) the answer to the last question will NOT appear in any intermediate questions.

**Step 2: Duplication Control** Built by rule-based composition, our new dataset has considerably similar reasoning chains that have duplicate intermediate questions. To ensure the diversity and simplicity of our dataset, we filter out the reasoning chains by a preset duplication degree which is defined by the number of questions that co-existed in other chains within the same reasoning type.

**Step 3: Binary Question Generation** We notice that MHQA datasets also include general interrogative questions which should be answered by "Yes" or "No", rather than a named entity. Therefore, we leverage the LLM to reform the last QA $(q_n, a_n)$ of some reasoning chains to binary question with 4 manually designed in-context demonstrations. For each reasoning type, we randomly sample 10% reasoning chains for positive question generation and

Table 1: Comparison of different approaches on four MHQA benchmarks. The fine-tuning methods are fine-tuned on the train split of NQ (Kwiatkowski et al., 2019) dataset. Among them, methods marked with "†" use the Wikipedia dump (Karpukhin et al., 2020) as extra corpus. For retrieval-based methods, we use a fine-tuned DPR (Karpukhin et al., 2020) to retrieve top-5 documents from Wikipedia as context and employ LLM as Reader to answer the question based on the context. Methods based on ChatGPT are performed by `gpt-3.5-turbo-0301` version.

| Methods | MSQ | | HotpotQA | | 2Wiki | | CWebQ | | Average | |
|---|---|---|---|---|---|---|---|---|---|---|
| | EM | F1 | EM | F1 | EM | F1 | EM | F1 | EM | F1 |
| *Fine-tuning methods with extra corpus* | | | | | | | | | | |
| DPR† (Karpukhin et al., 2020) | 7.5 | 16.6 | 14.7 | 23.0 | 6.5 | 14.6 | 21.3 | 30.2 | 12.5 | 21.1 |
| RAG† (Lewis et al., 2020) | 3.9 | 8.2 | 9.6 | 15.5 | 15.7 | 18.6 | 13.3 | 15.8 | 10.6 | 14.5 |
| REALM† (Guu et al., 2020) | 3.7 | 8.5 | 15.3 | 22.0 | 5.1 | 9.6 | 21.1 | 27.3 | 11.3 | 16.9 |
| T5-11B-SSM (Roberts et al., 2020) | 10.6 | 16.6 | 15.4 | 22.4 | 15.6 | 20.5 | 28.5 | 35.5 | 17.5 | 23.8 |
| *Retrieval-based methods with LLMs* | | | | | | | | | | |
| DPR† + ChatGPT | 1.7 | 4.1 | 15.8 | 21.4 | 10.9 | 18.1 | 13.7 | 18.7 | 10.5 | 15.6 |
| DPR† + Alpaca-13B (Taori et al., 2023) | 2.2 | 8.4 | 12.0 | 21.5 | 12.6 | 21.2 | 15.9 | 27.1 | 10.7 | 19.5 |
| DPR† + Vicuna-13B (Chiang et al., 2023) | 2.2 | 7.4 | 18.6 | 25.8 | 23.9 | 27.6 | 20.3 | 27.9 | 16.2 | 22.1 |
| DPR† + WizardLM-13B (Xu et al., 2023) | 3.5 | 10.0 | 19.9 | 28.4 | 22.8 | 27.5 | 24.2 | 32.2 | 17.6 | 24.5 |
| DPR† + InstructGPT (Ouyang et al., 2022) | 4.8 | 11.6 | 26.3 | 34.8 | 23.3 | 27.1 | 34.4 | 41.6 | 22.2 | 28.8 |
| *LLM-only methods on ChatGPT* | | | | | | | | | | |
| Zero-Shot | 3.1 | 7.3 | 22.4 | 30.0 | 18.7 | 21.7 | 31.6 | 37.5 | 19.0 | 24.1 |
| Self-Prompting (Li et al., 2022) | 2.9 | 6.2 | 23.8 | 31.2 | 18.9 | 23.5 | 26.8 | 32.6 | 18.1 | 23.4 |
| GENREAD (Yu et al., 2023) | 8.6 | 14.6 | **33.2** | 42.6 | **30.4** | **35.3** | 33.7 | 40.1 | 26.5 | 33.2 |
| Zero-shot-CoT (Kojima et al., 2023) | 5.0 | 8.8 | 22.6 | 29.6 | 24.3 | 27.1 | 30.3 | 36.2 | 20.6 | 25.4 |
| Auto-CoT (Zhang et al., 2022) | 8.1 | 13.6 | 26.1 | 36.3 | 26.2 | 30.2 | 29.9 | 38.4 | 22.6 | 29.6 |
| Manual-CoT (random) (Wei et al., 2023) | 12.3 | 19.2 | 32.4 | **43.7** | 27.7 | 34.6 | 36.6 | 43.0 | 27.3 | 35.1 |
| SP-CoT *(Ours)* | **14.5** | **22.6** | **33.2** | 42.9 | 30.1 | 34.7 | **37.5** | **43.6** | **28.8** | **36.0** |

10% for negative ones. Then we reform a new reasoning chain by the generated binary question together with its previous question hops and add it to the dataset.

**Step 4: Multi-Hop Question Generation** Now we need to generate multi-hop questions, to which the previously generated question chains serve as their intermediate reasoning steps. For each question chain, we iteratively replace the answer $a_i$ to an intermediate question $q_i$ in the next-hop question $q_{i+1}$ by $[q_i]$ until the last question $q_n$ is replaced, which indicates a relative clause. Then we leverage the LLM to reform it into a natural multi-hop question with 4 manually designed in-context demonstrations.

After the pipeline above, we construct a high quality ODMR dataset with 2-4 hops, including the overall multi-hop question, the decomposed reasoning chains with detailed QA quadruplets. With the double-check in generation and the composability criteria, we automatically build a high quality new dataset. Detailed prompt templates are presented in Figure 3 and Appendix B.

### 3.3 Adaptive In-context Demonstration

In this stage, we sample multi-hop questions from our generated ODMR dataset as in-context demonstrations.

**Clustering-based Retrieval** Some previous works (Zhang et al., 2022; Li et al., 2022) have shown that clustering-based methods benefit from the diversity of demonstrations. We adopt a clustering-based retrieval approach to adaptively sample in-context demonstrations for the input question. First, all the questions are projected to a high dimension hidden space by encoding with Sentence-BERT (Reimers and Gurevych, 2019). Suppose we need $n$ in-context demonstrations. Given a test question $Q$, we use *k-means* to cluster the question embeddings into $n$ clusters and adaptively retrieve the question with the highest cosine similarity to $Q$ from each cluster.

**Build Reasoning Chain** For each sampled example, we sequentially concatenate the explanation from each hop, prefaced by `"Step {i}:"`, to construct a reasoning chain.

## 4 Experiments

Our research questions (RQs) are:

RQ1: *To what extent can SP-CoT boost the LLMs on our four ODMR benchmarks, compared with other LLM-only methods?*

RQ2: *Is SP-CoT generally effective on recent popular instruction-following LLMs?*

To this end, we conduct experiments on four MHQA datasets that require complex multi-step reasoning and compare different methods across

Table 2: The performance (EM) of our method on recent popular LLMs. We use `text-davinci-003` for InstructGPT. On small-scale LLMs, SP-CoT nearly doubles the average zero-shot performance on four MHQA benchmarks.

| Model | Size | Method | MSQ | HotpotQA | 2Wiki | CWebQ | Mean | Boost |
|---|---|---|---|---|---|---|---|---|
| Alpaca (Taori et al., 2023) | 13B | Zero-shot | 0.9 | 9.5 | 12.5 | 12.6 | 8.9 | - |
| Alpaca (Taori et al., 2023) | 13B | SP-CoT | 5.3 | 19.8 | 17.8 | 26.5 | 17.4 | 8.5↑ |
| Vicuna (Chiang et al., 2023) | 13B | Zero-shot | 2.3 | 11.5 | 18.1 | 18.3 | 12.6 | - |
| Vicuna (Chiang et al., 2023) | 13B | SP-CoT | 8.5 | 23.3 | 22.0 | 32.2 | 21.5 | 8.9↑ |
| WizardLM (Xu et al., 2023) | 13B | Zero-shot | 2.0 | 10.3 | 13.6 | 17.5 | 10.9 | - |
| WizardLM (Xu et al., 2023) | 13B | SP-CoT | 7.3 | 24.0 | 27.2 | 31.3 | 22.5 | 11.6↑ |
| InstructGPT (Ouyang et al., 2022) | 175B | Zero-shot | 4.4 | 25.6 | 22.8 | 38.6 | 22.9 | - |
| InstructGPT (Ouyang et al., 2022) | 175B | SP-CoT | 14.8 | 37.4 | 33.7 | 46.6 | 33.1 | 10.2↑ |

different LLMs.

## 4.1 Benchmarks and Evaluation Metrics

We choose the following four MHQA datasets: CWebQ, HotpotQA, 2Wiki and MSQ. We set them as ODMR benchmarks by taking only the question and the answer in each example. Dataset introduction and statistics are detailed in Appendix A.

We adopt the exact match (EM) and F1 scores as our evaluation metrics. Based on the evaluation script of Karpukhin et al. (2020), we add a preprocessing step which ignores the content within "()" and splits the answer strings by certain delimiters to extract multiple answers.

## 4.2 Experiment Settings

For reference, we experiment with fine-tuning methods using an extra corpus, which are fine-tuned on the training split of NQ (Kwiatkowski et al., 2019) dataset and most of them adopt the Wikipedia dump (Karpukhin et al., 2020) as extra corpus. We also test our implementation of the retrieval-based methods on most recent LLMs for reference. Specifically, we use a fine-tuned DPR (Karpukhin et al., 2020) to retrieve top-5 documents from Wikipedia as context and employ LLM as Reader to answer the question based on the context. Detailed prompt templates and parameter settings are provided in the Appendix B.

Unless otherwise specified, we use Sentence-BERT (`all-mpnet-base-v2`) for question encoding following previous works. The default number of in-context demonstrations is 8 and the demonstrations are sampled by the maximum cosine similarity of questions in each cluster.

For RQ1, we adopt ChatGPT (`gpt-3.5-turbo-0301`) as the LLM to conduct the following experiments. According to OpenAI,[3] `gpt-3.5-turbo-0301` is an improvement on the InstructGPT `text-`

³https://platform.openai.com

davinci-003 model, which performs at a similar capability level to `text-davinci-003` for inference. We use the whole development set of each dataset in our experiments.

For RQ2, we not only test InstructGPT (`text-davinci-003`), but also employ three smaller-scale (13B) LLMs: Alpaca (Taori et al., 2023), Vicuna (Chiang et al., 2023) and WizardLM (Xu et al., 2023), which are LLaMA (Touvron et al., 2023) models fine-tuned on different large-scale instruction-following datasets. To save computational cost, we conduct this experiment on subsets of the four datasets by randomly selecting 1000 samples from the test sets.

## 4.3 Experiment Results

The main results of RQ1 are shown in Table 1. Even with extra corpus, the models fine-tuned on NQ (Kwiatkowski et al., 2019) present poor performance due to the inherent challenges of MHQA. With the same Retriever model, the performance of retrieval-based methods depends largely on the LLM Readers. Compared to previous LLM-only works, our SP-CoT significantly outperforms the Auto-CoT by +6.2 EM and +6.4 F1 scores on average and surpasses the previous SOTA method GENREAD (Yu et al., 2023) by +2.3 EM and +2.8 F1 scores on average. On the most challenging benchmark MSQ, SP-CoT empowers ChatGPT to outperform other LLM-only methods by a decent margin.

We notice that SP-CoT significantly outperforms GENREAD on MSQ, confirming the effectiveness of providing high quality CoTs as in-context demonstrations for complex multi-hop questions. On the other three datasets, SP-CoT delivers comparable performance with GENREAD. However, GENREAD relies heavily on the generation faithfulness of LLMs, which is challenging for small-scale LLMs. By breaking down demanding instructions into step-by-step simple ones, our method

Table 3: The performance (EM) of different methods of demonstration selection. The results from random selection represent the mean value and standard deviation obtained from 3 runs, each with a different seed.

| Method | MSQ | HotpotQA | 2Wiki | CWebQ | Average |
|---|---|---|---|---|---|
| Random | $12.0_{\pm 0.8}$ | $29.5_{\pm 0.6}$ | $25.6_{\pm 1.2}$ | $34.3_{\pm 0.9}$ | $25.4_{\pm 0.3}$ |
| Retrieve | 10.5 | 27.9 | 24.3 | 33.5 | 24.1 |
| ClusterCenter | 10.4 | 26.2 | 22.8 | 33.0 | 23.1 |
| RetrieveInTypeCluster | 11.6 | 28.7 | 23.5 | **35.3** | 24.8 |
| RetrieveInCluster | 11.5 | **30.9** | **27.8** | 34.0 | **26.1** |

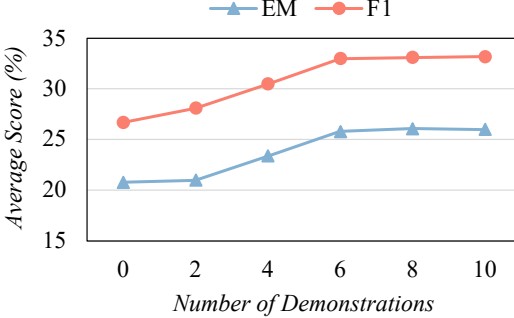

Figure 4: Average EM and F1 scores of different numbers of in-context demonstrations. The experiments are tested on 1k subsets of four ODMR benchmarks with ChatGPT (`gpt-3.5-turbo-0301`).

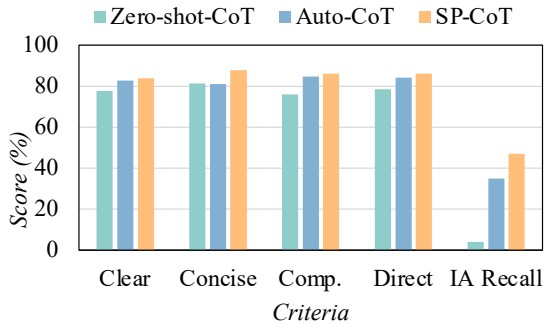

Figure 5: Evaluation results of the CoT generated by three methods. The first four scores are in terms of clearness, conciseness, comprehensibility (Comp.) and directness given by GPT-4 on 50 examples. The recall accuracy of intermediate answers (IA Recall) is reported on the questions that are correctly answered by all 3 methods.

is more applicable to small-scale LLMs, which is validated by Table 2.

Table 2 presents the results for RQ2. Our proposed SP-CoT proves to be generally effective by significantly boosting the performance of all these four LLMs on all four benchmarks. With SP-CoT, the performance of small-scale (13B) LLMs can be boosted to be on par with directly prompting LLMs that are over 10× larger, regardless of the elicited high quality intermediate reasoning steps.

## 5   Analysis

In this section, we explore the choices of the sampling methods and the number of demonstrations. Then we examine the quality of the intermediate reasoning steps elicited by SP-CoT and the quality of self-generation data. Unless otherwise specified, we use ChatGPT (`gpt-3.5-turbo-0301`) to conduct analysis on the same subsets mentioned in RQ2 settings.

### 5.1   Methods of Demonstration Sampling

The performance of ICL depends largely on the quality of demonstration sampling. We test the effectiveness of the following five strategies: randomly sampling (Random), sampling globally by maximum cosine similarity (Retrieve), sampling

the closest to the centroid in each cluster (ClusterCenter), sampling by the maximum cosine similarity in each cluster (RetrieveInCluster) and sampling the most similar QAs in each cluster in a certain reasoning type (RetrieveInTypeCluster). The reasoning type of the input question is determined by the most frequent reasoning type of its k-nearest neighbors. As indicated in Table 3, RetrieveInCluster (Li et al., 2022) is the best-performing strategy, which is exactly the strategy we adopt in previous experiments.

### 5.2   Impact of Demonstration Amount

Providing more in-context demonstrations empirically improves ICL performance; however, it also causes increasing computational cost. To this end, we investigate the trade-offs of number of demonstrations and the resulting performance boost. We report the EM and F1 scores over the four benchmarks for 2, 4, 6, 8, and 10 in-context demonstrations, as well as the scores in a zero-shot setting. As illustrated in Figure 4, the performance of SP-CoT increases with the number of demonstrations when the count is between 2 and 8; however, using 10 demonstrations doesn't yield any further

performance boost. In our main experiments, we opted for 8 as the default number of demonstrations, striking a balance between performance and cost.

### 5.3 Intermediate Reasoning Quality Analysis

Given the high-quality CoTs constructed by our proposed SP-CoT, we investigate the quality of intermediate reasoning steps generated during inference. For this analysis, we use the development set of MSQ, as it's the most challenging of the four datasets and offers decomposed step-by-step QAs. We compare the CoTs generated by Zero-shot-CoT, Auto-CoT and SP-CoT during inference. For fairness, we select 50 out of a total of 59 questions that all of the three methods answered correctly. First, we use GPT-4 to evaluate[4] the intermediate reasoning steps in terms of clearness, conciseness, comprehensibility and directness separately on a scale of 1 to 10. Additionally, we compute the recall accuracy of intermediate answers co-occurring in the reasoning steps of each method. For fairness, we only report the intermediate answer recall accuracy of correctly answered questions for each method. As depicted in Figure 5, GPT-4 highly favors our SP-CoT, which achieves nearly a 50% recall accuracy for intermediate answers. This suggests that SP-CoT elicits high-quality reasoning steps in terms of clearness, conciseness, comprehensibility, and directness.

### 6 Conclusion

In this work, we harness the capabilities of LLMs combined with self-prompted CoTs to tackle the intricate MHQA task within the open-domain context, termed as ODMR. Our innovative SP-CoT not only sets a new benchmark by surpassing preceding CoT prompting techniques but also outclasses the erstwhile SOTA LLM-only methodologies in open-domain question-answering. A distinguishing feature of SP-CoT is its proficiency in eliciting high-caliber intermediate reasoning steps, and its universal efficacy across both large and small-scale LLMs. We anticipate our innovative self-generation pipeline for ODMR to not just be foundational for SP-CoT, but also to pave the way for future research, catalyzing a shift towards leveraging self-generation in LLMs, by LLMs, and for LLMs.

---

[4]Script from https://github.com/lm-sys/FastChat and modified.

### Acknowledgements

This paper was partially supported by the Joint Research Project of Yangtze River Delta Science and Technology Innovation Community (No. 2022CSJGG1400) and under the technical support of National Key R&D Program of China (No. 2021YFC3340700). Our appreciation also extends to Ms. Yangyang Ding, who has been generous with her time and knowledge, offering constructive criticism and enriching discussions.

### Limitations

Our proposed method (SP-CoT) leverages the strong instruction-following power of LLMs. Such capability is easy to acquire through instruction fine-tuning for small-scale LLMs (even 7B), however, some LLMs proposed in early years may show poor capability in following human instructions due to lack of corresponding training before their release. Therefore, the performance of such LLMs may not be boosted by our proposed SP-CoT. In fact, we did not succeed in boosting the performance of GPT-NeoX by any of Zero-shot-CoT, Auto-CoT and SP-CoT. GPT-NeoX is a 20B LLMs released in early 2022, which shows poor instruction-following capability. Please note that neither GENREAD (Yu et al., 2023) nor Self-prompting (Li et al., 2022) boosts the performance of GPT-NeoX.

It is acknowledged that the efficacy of LLM-only approaches is predominantly reliant on the LLMs themselves. With smaller-scale LLMs, specifically those of 13B scale, our SP-CoT together with other CoT methodologies, demonstrate comparable or similar performance enhancement across four ODMR benchmarks as presented in Table 6. The consistent performance of handcrafted CoTs remains ambivalent across different LLMs and benchmarks; our empirical observations indicate that Manual-CoT occasionally outperforms SP-CoT, while at other instances, it does not.

Given the potential for LLMs to generate imprecise information, the process by which our SP-CoT produces datasets might also result in the emergence of inaccurate QA pairs as well as erroneous explanations. Despite the incorporation of a double-check mechanism to ensure data integrity, certain errors and inaccuracies are inevitably present.

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

# A Datasets

## A.1 Introduction

**HotpotQA (Yang et al., 2018)** HotpotQA is a widely used dataset for multi-hop question-answering (MHQA), which contains 113k multi-hop questions in natural language. The questions are collected by crowdsourcing based on Wikipedia articles with human annotated supporting evidence and answers.

**2WikiMultiHopQA (Ho et al., 2020)** 2Wiki-MultiHopQA s a recently proposed large-scale MHQA dataset, which contains over 192k samples constructed jointly from Wikipedia and Wikidata.

**MuSiQue-Ans (Trivedi et al., 2022)** MuSiQue-Ans (MSQ) is a recent challenging MHQA dataset created via single-hop question composition. It includes 25k 2-4 hop questions with six different composition structures. Although MSQ is composed from existing datasets, it poses $3\times$ the human-machine gap with a substantially lower disconnected reasoning score.

**ComplexWebQuestions (Talmor and Berant, 2018)** ComplexWebQuestions is a manually generated MHQA dataset of 35k QA pairs. CWebQ is generated by rephrasing questions generated by machine from existing dataset.

## A.2 Statistics

The statistics of four datasets are shown in Table 4.

Table 4: Statistics of development set of four MHQA benchmarks, average tokens for questions, answers and average number of reasoning steps. CWebQ does not provide supporting evidence or question decomposition.

|  | MSQ | HotpotQA | CWebQ | 2Wiki |
|---|---|---|---|---|
| Q len. | 18.11 | 15.83 | 13.37 | 11.98 |
| A len. | 2.8 | 2.46 | 2.42 | 2.41 |
| Steps | 2.65 | 2.68 | - | 2.47 |
| Size | 2417 | 7405 | 3519 | 12576 |

## B Prompt Templates

### B.1 First-Hop QA Generation

We following the notations described in Section 3. The templates are:

1. Name {Number} {Topic}:

2. Generate a Wikipedia passage about $\{k_1\}$.

3. Passage about $\{k_1\}$:\n$\{p_1\}$\n\nGenerate a question to which the answer is the entity $\{a_1\}$.

4. Passage about $\{k_1\}$:\n$\{p_1\}$\n\nQuestion:\n$\{q_1\}$\n\nExtract the answer directly from the passage in less words as possible.

5. Passage about $\{k_1\}$:\n$\{p_1\}$\n\n Question:\n$\{q_1\}$\n\nAnswer:\n$\{a_1\}$\n\nAccording to an evidence from the passage to support the answer, rewrite it to make its meaning clear without passage.

### B.2 Second-Hop QA Generation

1. Generate a Wikipedia passage about $\{k_2\}$.

2. Passage about $\{k_2\}$:\n$\{p_2\}$\n\nGenerate a question that meets the following conditions: 1. contains the term '$\{k_2\}$' in question, 2. the answer is $\{a_2\}$, 3. Avoid the following entities in the question: $\{k_2\}$

3. Passage about $\{k_2\}$:\n$\{p_2\}$\n\nQuestion:\n$\{q_2\}$\n\nExtract the answer directly from the passage in less words as possible.

4. Passage about $\{k_2\}$:\n$\{p_2\}$\n\n Question:\n$\{q_1\}$\n\nAnswer:\n$\{a_2\}$\n\nAccording to an evidence from the passage to support the answer, rewrite it to make its meaning clear without passage.

### B.3 Binary Question Generation

- Question: $\{q_n\}$\nAnswer: $\{a_n\}$\nReform the question to a general interrogative question that can be answered with yes:

- Question: $\{q_n\}$\nAnswer: $\{a_n\}$\nReform the question to a general interrogative question that can be answered with no:

### B.4 Multi-Hop Question Generation

- Raw question: $\{q_n\}$\nReplace the sentence within [] with a relative clause and make the raw question into a natural question:

### B.5 CoT Construction

Suppose $q^*$ is the generated multi-hop question, $e_i$ denotes the explanation from intermediate hop ($q_i$, $a_i$, $e_i$), $a^*$ is the answer of the last hop ($a^* = a_n$). The template is:

- Question: $\{q^*\}$\nAnswer: Let's think step by step:\nStep 1: $\{e_1\}$\nStep 2: $\{e_2\}$\n ... Therefore, the answer in just one entity is: $\{a^*\}$,

### B.6 Inference

#### B.6.1 Zero-shot

Given a question $Q$, the inference template is:

- Answer the following question with just one entity:\nQuestion: $\{Q\}$\nAnswer:

#### B.6.2 SP-CoT

Suppose we have the input question $Q$ and 2 demonstrations ($q_1$, $r_1$, $a_1$), ($q_2$, $r_2$, $a_2$), where $q_i$, $r_i$, $a_i$ denote the question, CoT and answer of the $i^{th}$ demonstration. The inference template is:

- Question: $\{q_1\}$\n$\{r_1\}$\n\nQuestion: $\{q_2\}$\n$\{r_2\}$\n\nQuestion: $\{Q\}$\nAnswer: Let's think step by step:\n

## C Experiment Settings

### C.1 Hyperparameters

This section is the experiment settings on ChatGPT (gpt-3.5-turbo-0301) only, for more settings of other LLMs used in our experiments, please see our code. Our code and datasets are publicly available at https://github.com/noewangjy/SP-CoT.

Table 5: Performance (EM/F1) of additional CoT variants. In our experiment, Manual-CoT (Cherry-Pick) adopts 8 cherry-picked questions and their CoTs manually written by the authors. The results of Manual-CoT (Random) report the mean EM scores of randomly selected questions and theirs manual CoTs for 2 experiments across the 4 benchmarks. Self-Consistency (Wang et al., 2023) is based on SP-CoT with 5 responses for each question.

| Method | MSQ | HotpotQA | 2Wiki | CWebQ | Average |
|---|---|---|---|---|---|
| Zero-shot | 3.1/7.3 | 22.4/30.0 | 18.7/21.7 | 31.6/37.5 | 19.0/24.1 |
| Zero-shot-CoT (Kojima et al., 2023) | 5.0/8.8 | 22.6/29.6 | 24.3/27.1 | 30.3/36.2 | 20.6/25.4 |
| Manual-CoT (Wei et al., 2023) (Random) | 12.3/19.2 | 32.4/43.7 | 27.7/34.6 | 36.6/43.0 | 27.3/35.1 |
| Manual-CoT (Wei et al., 2023) (Cherry-Pick) | 13.7/21.9 | 33.9/44.7 | 31.5/38.6 | 37.2/44.0 | 29.1/37.3 |
| Auto-CoT (Zhang et al., 2022) | 8.1/13.6 | 26.1/36.3 | 26.2/30.2 | 29.9/38.3 | 22.6/29.6 |
| SP-CoT (Ours) | 14.5/22.6 | 33.2/42.9 | 30.1/34.7 | 37.5/43.6 | 28.8/36.0 |
| SP-CoT + Self-Consistency (Wang et al., 2023) | 18.3/28.3 | -/- | -/- | 47.1/54.0 | -/- |

Table 6: The performance (EM) of CoT methods with recent popular LLMs on 1k subsets of test sets. We use `gpt-3.5-turbo-0301` for ChatGPT and `text-davinci-003` for InstructGPT. On smaller-scale (13B) LLMs, CoT methods achieve comparable performance boost on four MHQA benchmarks.

| Method | ChatGPT | InstructGPT | Alpaca-13B | Vicuna-13B | Wizard-13B |
|---|---|---|---|---|---|
| Zero-shot | 20.8 | 22.9 | 8.9 | 12.6 | 10.9 |
| Manual-CoT (Wei et al., 2023) (Cherry-Pick) | 28.7 | 33.1 | 18.0 | 23.7 | 24.7 |
| Manual-CoT (Wei et al., 2023) (Random) | 26.3 | 32.4 | 17.7 | 22.0 | 24.0 |
| Auto-CoT (Zhang et al., 2022) | 21.2 | 31.4 | 16.9 | 21.3 | 23.3 |
| SP-CoT (Ours) | 26.1 | 33.1 | 17.4 | 21.5 | 22.5 |

### C.1.1 System message

*"You should use your knowledge to answer the question to the best of your ability, not refuse to answer, even though you know your knowledge is sometimes out of date. If some references are uncertain, answer all possible cases rather than requesting further information."*

### C.1.2 Temperature

In most cases, the default temperature is set to 0 for obtaining a most deterministic response. When we ask ChatGPT to name some terms or to generation a question, the temperature is set to 1.0 for more diversity.

### C.2 LLMs

The 13B LLMs used in our experiments are from Huggingface Hub, use `chavinlo/gpt4-x-alpaca` for Alpaca-13B, `TheBloke/wizard-vicuna-13B-HF` for Vicuna-13B and `TheBloke/wizardLM-13B-1.0-fp16` for WizardLM-13B.

## D Additional Experiments

### D.1 Comparison of CoT Variants

To provide a more comprehensive picture of current CoT methods on ODQA, we report hereby (Table 5) the performance of additional CoT variants, including Manual-CoT (Wei et al., 2023) and Auto-CoT (Zhang et al., 2022) on ChatGPT (`gpt-3.5-turbo-0301`).

In our experiment, Manual-CoT (Cherry-Pick) adopts 8 cherry-picked questions and their CoTs manually writen by the authors. The results of Manual-CoT (Random) report the mean EM scores of randomly selected questions and theirs manual CoTs for 2 experiments across the 4 benchmarks. Self-Consistency (Wang et al., 2023) is based on SP-CoT with 5 responses for each question.

To the best of our knowledge, Self-Consistency (Wang et al., 2023) is orthogonal to existing CoT methods, including SP-CoT. Although Self-Consistency boosts the performance of SP-CoT to a higher level (10%-30% increase for 5 runs), it's worth noting that the cost of Self-Consistency is also 5 times higher.

In Table 6, we report the performance (EM) of CoT methods with recent popular LLMs on 1k subsets of the test sets. The scores are the average EM scores on 4 ODMR benchmarks. Although Manual-CoT (Wei et al., 2023) outperforms Automated methods, it requires high quality human-labeled CoTs, which is not always accessible in real world applications. Since the cherry-picked CoTs take the dataset features in to consideration, we consider their results as the **theoretical upper limit** of automated approaches. Compared to previously automatic SOTA method (Auto-CoT), our proposed SP-CoT shows a decent performance boost in most cases.

Table 7: The scale and average hops of each ODMR dataset generated in this paper. Please note that the scale and the average hops are largely decided by the self-generation setting and the duplication control process.

| | ChatGPT | Alpaca-13B | Vicuna-13B | WizardLM-13B |
|---|---|---|---|---|
| Number of Samples | 3550 | 1354 | 1562 | 1604 |
| Avg. Number of Hops | 3.04 | 2.82 | 2.92 | 3.16 |
| Avg. Question tokens | 34.61 | 25.35 | 31.05 | 34.30 |
| Avg. Answer tokens | 1.86 | 1.99 | 2.02 | 1.91 |

## D.2 SP-CoT on GrailQA

We report our experiment of CoT methods on 1k subset of test set provided by GrailQA (Gu et al., 2021). According to our ODMR setting, no external knowledge is provided to LLMs. From the results below, we notice that our proposed SP-CoT is effective on GrailQA, our results on InstructGPT (`text-davinci-003`) are presented in Table 8.

Table 8: Performance of CoT methods on 1k subset of test set provided by GrailQA (Gu et al., 2021).

| Method | EM | F1 |
|---|---|---|
| Zero-shot | 12.9 | 24.3 |
| Zero-shot-CoT (Kojima et al., 2023) | 13.5 | 25.2 |
| Manual-CoT (Wei et al., 2023) (Random) | 17.7 | 30.7 |
| SP-CoT (Ours) | 16.0 | 28.2 |

## E Constructed ODMR Datasets

### E.1 Overview and scale

For better understanding of the constructed ODMR datasets, we offer a well-designed figure (Figure 6) to illustrate the six types of generated questions and their step-by-step decomposition. The scale of the generated ODMR datasets is about 1-4k (Table 7), however, it's largely dependent by the self-generation setting (how many examples to generate?) and the Duplication Control process in Stage 2 Step 2 (How many examples to keep?) To be more specific, the number of topic terms for self-generation decides the scale of generated 2-hop question pairs, and the level of duplication (how many existing question hops are allowed when constructing a new reasoning chain) decides the scale of the remaining examples after filtering.

### E.2 Topics

The 29 manually designed topics for generation are: *politicians, athletes, sports teams, sports events, countries, cities, historical figures, historical events, wars, religions, singers, songs, actors or actresses, movies or TV series, writers, books, painters, paintings, composers, classical music, tourist attractions, scientists, scientific terms, video games, animals, plants, foods, enterprises, international organizations.*

### E.3 Quality Control

To ensure the self-generation quality, two mechanisms are included in our proposed method.

**Self-validation in self-generation**: To ensure the quality of generated QA pairs, we employ a double-check process, where we demand the LLM to answer the generated question given the generated context and double-check If the generated answer is accordant with the target answer.

**Composability criteria in composition**: Two single-hop QA pairs $(q_1, a_1)$ and $(q_2, a_2)$ are composable into a multi-hop question $Q$ with $a_2$ as a valid answer if $a_1$ is a named entity and it is mentioned in $q_2$. Such criteria are already satisfied when our 2-hop QA pairs are generated, we use it for connecting more questions.

### E.4 Composition rather than Generation

Directly generating k-hop questions will produce many highly-duplicated reasoning chains, which is less effective than conposition with 2-hop QAs.

Take a connected 3-hop QAs as example: $(Q_1, A_1) \rightarrow (Q_2, A_2) \rightarrow (Q_3, A_3)$, where $A_1$ in $Q_2$, $A_2$ in $Q_3$.

Suppose there are 2 valid next-hop QAs $(Q_4, A_4)$ and $(Q_5, A_5)$ for $(Q_3, A_3)$. Now we have 2 generated 4-hop reasoning chains: $(Q_1, A_1) \rightarrow (Q_2, A_2) \rightarrow (Q_3, A_3) \rightarrow (Q_4, A_4)$ and $(Q_1, A_1) \rightarrow (Q_2, A_2) \rightarrow (Q_3, A_3) \rightarrow (Q_5, A_5)$

which are **highly-duplicated**. When directly generating k-hop reasoning chains, the number of highly-duplicated chains will increase exponentially and such chains will be filtered out in the Duplication Control Process (Stage 2, Step 2).

Furthermore, when there are more that 2 question hops in one reasoning chain, more effort should be made to ensure direct acyclic graphs (DAGs). An example of cyclic reasoning chain is $(Q_1, A_1) \rightarrow (Q_2, A_2) \rightarrow (Q_1, A_1)$, which should be avoided.

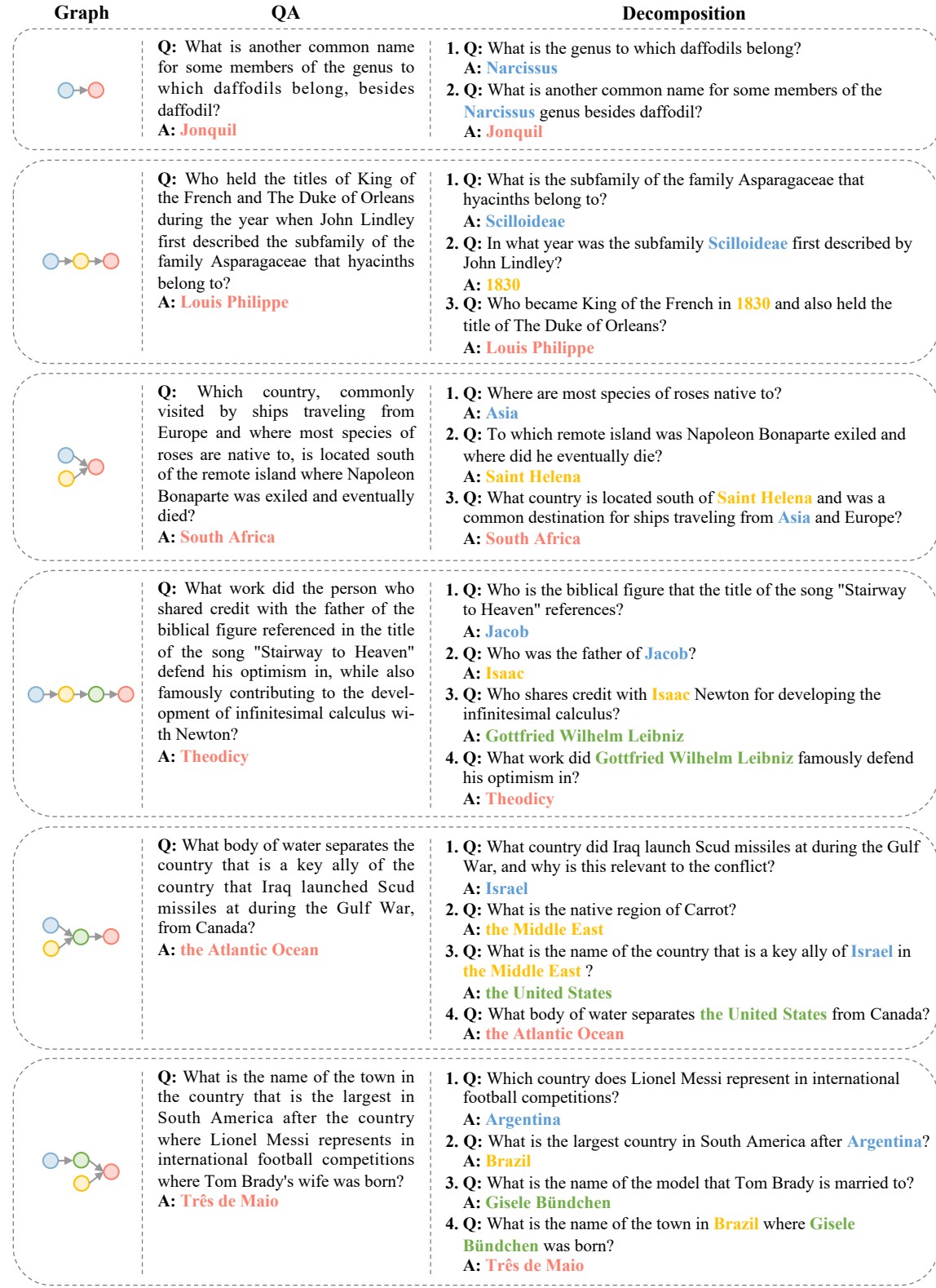

| Graph | QA | Decomposition |
|---|---|---|

**Q:** What is another common name for some members of the genus to which daffodils belong, besides daffodil?
**A:** Jonquil

**1. Q:** What is the genus to which daffodils belong?
**A:** Narcissus
**2. Q:** What is another common name for some members of the Narcissus genus besides daffodil?
**A:** Jonquil

**Q:** Who held the titles of King of the French and The Duke of Orleans during the year when John Lindley first described the subfamily of the family Asparagaceae that hyacinths belong to?
**A:** Louis Philippe

**1. Q:** What is the subfamily of the family Asparagaceae that hyacinths belong to?
**A:** Scilloideae
**2. Q:** In what year was the subfamily Scilloideae first described by John Lindley?
**A:** 1830
**3. Q:** Who became King of the French in 1830 and also held the title of The Duke of Orleans?
**A:** Louis Philippe

**Q:** Which country, commonly visited by ships traveling from Europe and where most species of roses are native to, is located south of the remote island where Napoleon Bonaparte was exiled and eventually died?
**A:** South Africa

**1. Q:** Where are most species of roses native to?
**A:** Asia
**2. Q:** To which remote island was Napoleon Bonaparte exiled and where did he eventually die?
**A:** Saint Helena
**3. Q:** What country is located south of Saint Helena and was a common destination for ships traveling from Asia and Europe?
**A:** South Africa

**Q:** What work did the person who shared credit with the father of the biblical figure referenced in the title of the song "Stairway to Heaven" defend his optimism in, while also famously contributing to the development of infinitesimal calculus with Newton?
**A:** Theodicy

**1. Q:** Who is the biblical figure that the title of the song "Stairway to Heaven" references?
**A:** Jacob
**2. Q:** Who was the father of Jacob?
**A:** Isaac
**3. Q:** Who shares credit with Isaac Newton for developing the infinitesimal calculus?
**A:** Gottfried Wilhelm Leibniz
**4. Q:** What work did Gottfried Wilhelm Leibniz famously defend his optimism in?
**A:** Theodicy

**Q:** What body of water separates the country that is a key ally of the country that Iraq launched Scud missiles at during the Gulf War, from Canada?
**A:** the Atlantic Ocean

**1. Q:** What country did Iraq launch Scud missiles at during the Gulf War, and why is this relevant to the conflict?
**A:** Israel
**2. Q:** What is the native region of Carrot?
**A:** the Middle East
**3. Q:** What is the name of the country that is a key ally of Israel in the Middle East ?
**A:** the United States
**4. Q:** What body of water separates the United States from Canada?
**A:** the Atlantic Ocean

**Q:** What is the name of the town in the country that is the largest in South America after the country where Lionel Messi represents in international football competitions where Tom Brady's wife was born?
**A:** Três de Maio

**1. Q:** Which country does Lionel Messi represent in international football competitions?
**A:** Argentina
**2. Q:** What is the largest country in South America after Argentina?
**A:** Brazil
**3. Q:** What is the name of the model that Tom Brady is married to?
**A:** Gisele Bündchen
**4. Q:** What is the name of the town in Brazil where Gisele Bündchen was born?
**A:** Três de Maio

Figure 6: The illustration of six reasoning types in our automated dataset construction pipeline. These selected examples are self-generated by ChatGPT (gpt-3.5-turbo-0301), Vicuna-13B and WizardLM-13B.