# OpenReview forum: "Self-prompted Chain-of-Thought on Large Language Models for Open-domain Multi-hop Reasoning"
_EMNLP/2023/Conference — EMNLP 2023 Findings_

### Official Review · Reviewer_mWKK · 2023-08-04

**Soundness:** 3

**Excitement:**

4: Strong: This paper deepens the understanding of some phenomenon or lowers the barriers to an existing research direction.

**Missing References:**

[1] Wang et al., Self-Consistency Improves Chain of Thought Reasoning in Language Models. ICLR 2023

[2] Fu et al., Complexity-Based Prompting for Multi-step Reasoning. ICLR 2023

[3] Zhang et al., Automatic Chain of Thought Prompting in Large Language Models. ICLR 2023.

**Paper Topic And Main Contributions:**

The authors propose Self-prompted Chain-of-Thought (SP-CoT) to automatically produce high quality CoT of LLMs for Open-domain Question-Answering (ODQA) task. Specifically, the process is divided into 2 stages (1) Generating 2-hop commonsense quadruplets, (2) Connecting 2-hop QA quadruplets to construct multi-hop reasoning chains. The proposed method are empirically demonstrated to yield promising improvements over 4 benchmarks across different QA datsets. The quality of generated CoT is also extensively evaluated.

**Questions For The Authors:**

A. Is it possible to self-generate 4-hop or k-hop QAs directly rather than 2-hop QAs with composition? Some additional rationales together with potential empirical demonstrations of  the effectiveness of the proposed 2-hop QAs  with composition would be beneficial.

**Reasons To Accept:**

The work is well-written and presents clear motivations.

The automated pipeline to generate ODQA datasets can be promising for more complex NLP tasks.

Strong performance improvements over zero-shot inference across different benchmarks and datasets.

**Reasons To Reject:**

Reported performance of additional CoT variants [1,2,3] (Table 1 and 2) would be beneficial to observe the complete picture of current CoT methods on ODQA tasks.


[1] Wang et al., Self-Consistency Improves Chain of Thought Reasoning in Language Models. ICLR 2023

[2] Fu et al., Complexity-Based Prompting for Multi-step Reasoning. ICLR 2023

[3] Zhang et al., Automatic Chain of Thought Prompting in Large Language Models. ICLR 2023.

**Reproducibility:**

3: Could reproduce the results with some difficulty. The settings of parameters are underspecified or subjectively determined; the training/evaluation data are not widely available.

**Reviewer Confidence:**

2: Willing to defend my evaluation, but it is fairly likely that I missed some details, didn't understand some central points, or can't be sure about the novelty of the work.

**Typos Grammar Style And Presentation Improvements:**

Line 421: sampling -> sample

Line 499: shows-> show

Referring to specific sections of the Appendix is recommended (Line 308, 443, etc.)

---

> ### Author Rebuttal · Authors · 2023-08-29
>
> Thank you for taking the time to review our paper and for your thoughtful feedback.
>
>
>
> > 1. **Reported performance of additional CoT variants [1,2,3] (Table 1 and 2) would be beneficial to observe the complete picture of current CoT methods on ODQA tasks.**
>
>
>
> To provide a more comprehensive picture of current CoT methods on ODQA, we report hereby the performance (EM scores) of additional CoT variants, including Manual-CoT [4] and Auto-CoT [3] on ChatGPT(`gpt-3.5-turbo-0301`).  In our experiment, Manual-CoT (Cherry-Pick) adopts 8 cherry-picked questions and their CoTs manually writen by the authors. The results of Manual-CoT (Random) report the mean EM scores of randomly selected questions and theirs manual CoTs for 2 experiments across the 4 benchmarks. Self-Consistency [1] is based on SP-CoT with 5 responses for each question.
>
> We report additional results for Table 1 to offer a more general overview of the performance of existing methods on our proposed ODMR task. The scores are EM/F1 on the full dev set of 4 ODMR benchmarks.
>
> |            Method            | MusiqueQA | HotpotQA  | WikiMHQA  |   CWebQ   |   Mean    |
> | :--------------------------: | :-------: | :-------: | :-------: | :-------: | :-------: |
> |           Zeroshot           |  3.1/7.3  | 22.4/30.0 | 18.7/21.7 | 31.6/37.5 | 19.0/24.1 |
> |       Zeroshot-CoT [5]       |  5.0/8.8  | 22.6/29.6 | 24.3/27.1 | 30.3/36.2 | 20.6/25.4 |
> |   Manual-CoT [4] (Random)    | 12.3/19.2 | 32.4/43.7 | 27.7/34.6 | 36.6/43.0 | 27.3/35.1 |
> | Manual-CoT [4] (Cherry-Pick) | 13.7/21.9 | 33.9/44.7 | 31.5/38.6 | 37.2/44.0 | 29.1/37.3 |
> |         Auto-CoT [3]         | 8.1/13.6  | 26.1/36.3 | 26.2/30.2 | 29.9/38.3 | 22.6/29.6 |
> |        SP-CoT (Ours)         | 14.5/22.6 | 33.2/42.9 | 30.1/34.7 | 37.5/43.6 | 28.8/36.0 |
> |     Self-Consistency [1]     | 18.3/28.3 |    -/-    |    -/-    | 47.1/54.0 |    -/-    |
>
> To the best of our knowledge, Self-Consistency [1] is orthogonal to existing CoT methods, including SP-CoT.  Although Self-Consistency boosts the performance of SP-CoT to a higher level (10%-30% increase for 5 runs), it’s worth noting that the cost of Self-Consistency is also 5 times higher.
>
> In Complexity-based prompting [2], the authors set up a complexity threshold for voting. While in our ODMR task, questions usually requires 2-4 hops, and 4-step reasoning is not better than 2-step reasoning for a 2-hop question. Therefore, reasoning complexity is independent of the performance in our cases, and  we cannot select a reasonable and practical complexity threshold.
>
> Although Manual-CoT [4] outperformes Automated methods, it requires high quality human-labeled CoTs, which is not always accessible in real world applications. Since the cherry-picked CoTs  take the dataset features in to consideration, we consider their results as the **theoretical  upper limit** of automated approaches. Compared to previously automatic SOTA method (Auto-CoT), our proposed SP-CoT shows a decent performance boost in most cases.
>
> Following are additional results of more LLMs on 1k subsets of the test sets. The scores are the average EM scores on 4 ODMR benchmarks.
>
> |            Method            | ChatGPT(`gpt-3.5-turbo-0301`) | InstructGPT(`text-davinci-003`) | Alpaca-13B | Vicuna-13B | Wizard-13B |
> | :--------------------------: | :---------------------------: | :-----------------------------: | :--------: | :--------: | :--------: |
> |           Zeroshot           |             20.8              |              22.9               |    8.9     |    12.6    |    10.9    |
> | Manual-CoT [4] (Cherry-Pick) |             28.7              |              33.1               |    18.0    |    23.7    |    24.7    |
> |   Manual-CoT [4] (Random)    |             26.3              |              32.4               |    17.7    |    22.0    |    24.0    |
> |         Auto-CoT [3]         |             21.2              |              31.4               |    16.9    |    21.3    |    23.3    |
> |        SP-CoT (Ours)         |             26.1              |              33.1               |    17.4    |    21.5    |    22.5    |
>
>
>
> > 2. **Is it possible to self-generate 4-hop or k-hop QAs directly rather than 2-hop QAs with composition? Some additional rationales together with potential empirical demonstrations of the effectiveness of the proposed 2-hop QAs with composition would be beneficial.**
>
> Directly generating k-hop questions will produce many highly-duplicated reasoning chains, which is less effective than conposition with 2-hop QAs.
>
> Let’s take a connected 3-hop QAs as example: `(Q1, A1) -> (Q2, A2) -> (Q3, A3)`, where `A1` in `Q2`, `A2` in `Q3`.
>
> Suppose there are 2 valid next-hop QAs `(Q4, A4)` and `(Q5, A5)` for `(Q3, A3)`.
>
> Now we have 2 generated 4-hop reasoning chains:
>
> - `(Q1, A1) -> (Q2, A2) -> (Q3, A3) -> (Q4, A4)`
> - `(Q1, A1) -> (Q2, A2) -> (Q3, A3) -> (Q5, A5)`
>
> which are highly-duplicated. When directly generating k-hop reasoning chains, the number of highly-duplicated chains will increase exponentially and such chains will be filtered out in the Duplication Control Process (Stage 2, Step 2).
>
> Furthermore, when there are more that 2 question hops in one reasoning chain, more effort should be made to ensure direct acyclic graphs (DAGs). An example of cyclic reasoning chain is `(Q1, A1) -> (Q2, A2) -> (Q1, A1)`, which should be avoided.
>
>
>
> [1] Wang et al., Self-Consistency Improves Chain of Thought Reasoning in Language Models. ICLR 2023
>
> [2] Fu et al., Complexity-Based Prompting for Multi-step Reasoning. ICLR 2023
>
> [3] Zhang et al., Automatic Chain of Thought Prompting in Large Language Models. ICLR 2023.
>
> [4] Wei, J., Wang, X., Schuurmans, D., Bosma, M., Xia, F., Chi, E., ... & Zhou, D. (2022). Chain-of-thought prompting elicits reasoning in large language models. *Advances in Neural Information Processing Systems*, *35*, 24824-24837.
>
> [5] Kojima, T., Gu, S. S., Reid, M., Matsuo, Y., & Iwasawa, Y. (2022). Large language models are zero-shot reasoners. *Advances in neural information processing systems*, *35*, 22199-22213.

---

### Official Review · Reviewer_K9QY · 2023-08-05

**Soundness:** 3

**Excitement:**

3: Ambivalent: It has merits (e.g., it reports state-of-the-art results, the idea is nice), but there are key weaknesses (e.g., it describes incremental work), and it can significantly benefit from another round of revision. However, I won't object to accepting it if my co-reviewers champion it.

**Paper Topic And Main Contributions:**

In this study, the authors utilize Large Language Models (LLMs) with self-prompted Chain-of-Thought (SP-CoT) to tackle the challenging tasks of Open-Domain Multi-Hop Reasoning (ODMR). Existing automated paradigm lacks of quality assurance while manual approaches suffer from limited scalability. The proposed process involves three stages. In stage 1, LLMs are prompted to iteratively generate 2-hop commonsense QA quadruplets containing context, question, answer, and explanation. In stage 2, these quadruplets are connected to form multi-hop reasoning chains, leading to the creation of an ODMR dataset through composition. In stage 3, a clustering-based sampling approach is adopted to dynamically select and construct in-context demonstrations for inference. Instead of extracting demonstrations from existing ODMR datasets, their method automatically generates demonstrations via LLMs. They evaluate the proposed method, which could consistently achieve SOTA results on MSQ, HotpotQA, 2Wiki and CWebQ datasets compared with the existing prompting methods.

**Questions For The Authors:**

1. In the Method section, in stage 1, the generated questions already include answers to intermediate questions. However, in stage 2, during step 4, why are the intermediate questions replaced with corresponding answers again?
2. It would be preferable if the examples presented in Figure 2 and Figure 3 in the paper could correspond to each other.
3. Recently, one of the commonly used datasets for Knowledge Base Question Answering (KBQA) is GrailQA, which also features multi-hop reasoning. I suggest to conduct an experiment on this dataset as well.


**Reasons To Accept:**

1.	The prompting method proposed for generating intermediate ODMR datasets is reasonable. Various details are taken into consideration, such as addressing the repetitiveness of intermediate questions and considering binary questions.

**Reasons To Reject:**

1.	The multi-hop question-answering (MHQA) methods in the baseline are only up to 2020, and some of the latest works on MHQA have not been mentioned, such as [3,4,5].
2.	More information of the constructed ODMR datasets should be introduced, which would be useful for readers to understand the quality of generated questions. For example, the scale of ODMR datasets and the process of determining the number of intermediate problems for each complex issue.
3.	Generating complex questions by composing intermediate sub-questions has been explored in traditional methods [1]. Selecting demonstrations via clustering-based methods has also been investigated by existing prompting methods for other tasks [2]. This weakens the innovation of the proposed methods. I suggest the authors to shed more light on the elaborate design of the proposed method for the ODMR tasks.

[1] Harsh Trivedi and Niranjan Balasubramanian and Tushar Khot and Ashish Sabharwal. MuSiQue: Multihop Questions via Single-hop Question Composition.
[2] Zhuosheng Zhang  and  Aston Zhang  and  Mu Li  and  Alex Smola. Automatic Chain of Thought Prompting in Large Language Models.
[3] Yingqi Qu and Yuchen Ding and Jing Liu and Kai Liu and Ruiyang Ren and Wayne Xin Zhao and Daxiang Dong and Hua Wu and Haifeng Wang. RocketQA: An Optimized Training Approach to Dense Passage Retrieval for Open-Domain Question Answering.
[4] Devendra Singh Sachan and Mostofa Patwary and Mohammad Shoeybi and Neel Kant and Wei Ping and William L Hamilton and Bryan Catanzaro. End-to-End Training of Neural Retrievers for Open-Domain Question Answering.
[5] Pratyay Banerjee and Chitta Baral. Knowledge Fusion and Semantic Knowledge Ranking for Open Domain Question Answering.


**Reproducibility:**

3: Could reproduce the results with some difficulty. The settings of parameters are underspecified or subjectively determined; the training/evaluation data are not widely available.

**Reviewer Confidence:**

1: Not my area, or paper was hard for me to understand. My evaluation is just an educated guess.

---

> ### Author Rebuttal · Authors · 2023-08-29
>
> Thanks for your feedback and suggestions, they are helping us to make our paper better. Here we are trying to make our idea more clear and easy to understand.
>
> > 1. **The multi-hop question-answering (MHQA) methods in the baseline are only up to 2020, and some of the latest works on MHQA have not been mentioned, such as [3,4,5].**
>
> Actually, our research scope is subject to a challenging and quite different setting,  multi-step question-answering without providing any external knowledge source. Different from existing MHQA works, our research focus on “**LLM-only**” for MHQA, which explores the capability of LLMs for multi-step reasoning via self-enhancement.
>
> The performance of retrieval-based models (DPR, RAG, REALM) are provided as reference results, rather than comparison baselines, to present the challenge of our newly proposed ODMR, which requires multi-step reasoning without external knowledge. To the best of our knowledge, reported ODQA models in this paper (DPR, RAG, REALM) have been the latest open-sourced models in consideration of our assumed knowledge-free setting in terms of our ODMR task. Such reference results could have been more recent if latest works [3,4,5] released open-sourced models.
>
> In the following table, we report more reference results on our proposed ODMR task, including latest knowledge-dependent methods [1,2] and our implementation of Retrieval-based methods on most recent LLMs (We use DPR to retrieve top-5 documents from Wikipedia as context and employ LLM as Reader to answer the question based on the context). The scores are EM/F1 on the full dev set of 4 ODMR benchmarks.
>
> |                            Method                            | MusiqueQA | HotpotQA  | WikiMHQA  |   CWebQ   |   Mean    |
> | :----------------------------------------------------------: | :-------: | :-------: | :-------: | :-------: | :-------: |
> |              DPR+ChatGPT (`gpt-3.5-turbo-0301`)              |  1.7/4.1  | 15.8/21.4 | 10.9/18.1 | 13.7/18.7 | 10.5/15.6 |
> |             DPR+InstructGPT (`text-davinci-003`)             | 4.8/11.6  | 26.3/34.8 | 23.3/27.1 | 34.4/41.6 | 22.2/28.8 |
> |                       DPR+Alpaca(13B)                        |  2.2/8.4  | 12.0/21.5 | 12.6/21.2 | 15.9/27.1 | 10.7/19.5 |
> |                       DPR+Vicuna(13B)                        |  2.2/7.4  | 18.6/25.8 | 23.9/27.6 | 20.3/27.9 | 16.2/22.1 |
> |                       DPR+Wizard(13B)                        | 3.5/10.0  | 19.9/28.4 | 22.8/27.5 | 24.2/32.2 | 17.6/24.5 |
> |                      IRCoT (GPT-3) [1]                       |  -/30.8   |  -/59.1   |  -/66.5   |    -/-    |    -/-    |
> | DEMONSTRATE–SEARCH–PREDICT (Codex(`code-davinci-002`)) [2] |    -/-    | 51.4/62.9 |    -/-    |    -/-    |    -/-    |
> |       SP-CoT (Ours, ChatGPT (`gpt-3.5-turbo-0301`))       | 14.5/22.6 | 33.2/42.9 | 30.1/34.7 | 37.5/43.6 | 28.8/36.0 |
>
>
>
> > 2. **More information of the constructed ODMR datasets should be introduced, which would be useful for readers to understand the quality of generated questions. For example, the scale of ODMR datasets and the process of determining the number of intermediate problems for each complex issue.**
>
> For better understanding of the constructed ODMR datasets, we offer a well-designed figure (Figure-6 in the Appendix, page 13) to illustrate the six types of generated questions and their step-by-step decomposition. The scale of the generated ODMR datasets is about 1-4k, however, it’s largely dependent by the self-generation setting (how many examples to generate?) and the Duplication Control process in Stage 2 Step 2 (How many examples to keep?) To be more specific, the number of topic terms for self-generation decides the scale of generated 2-hop question pairs, and the level of duplication (how many existing question hops are allowed when constructing a new reasoning chain) decides the scale of the remaining examples after filtering.
>
> To ensure the self-generation quality, two mechanisms are included in our proposed method.
>
> - **Self-validation in self-generation:** To ensure the quality of generated QA pairs, we employ a double-check process, where we demand the LLM to answer the generated question given the generated context and double-check if the generated answer is accordant with the target answer.
> - **Composability criteria in composition:** Two single-hop QA pairs $(q_1, a_1)$ and $(q_2, a_2)$ are composable into an multi-hop question $Q$ with $a_2$ as a valid answer if $a_1$ is a named entity and it is mentioned in $q_2$. Such criteria are already satisfied when our 2-hop QA pairs are generated, we use it for connecting more questions.
>
> The scale and average hops of each ODMR dataset generated in this paper are listed in the following table. Please note that the scale and the average hops are largely decided by the self-generation setting and the duplication control process.
>
> |          Model          | ChatGPT(`gpt-3.5-turbo-0301`) | Alpaca-13B | Vicuna-13B | Wizard-13B |
> | :---------------------: | :---------------------------: | :--------: | :--------: | :--------: |
> |   Number of  Samples    |             3550              |    1354    |    1562    |    1604    |
> |      Average Hops       |             3.04              |    2.82    |    2.92    |    3.16    |
> | Average Question tokens |             34.61             |   25.35    |   31.05    |   34.30    |
> |  Average Answer tokens  |             1.86              |    1.99    |    2.02    |    1.91    |
>
> The number of intermediate problems is determined during construction. For example, suppose we have 2 2-hop question pairs: `(Q1, A1)->(Q2, A2)` and `(Q3, A3) -> (Q4, A4)`. If `A2` appears in `Q3`, then it’s possible to construct a 4-hop reasoning chain (`Q1->Q2->Q3->Q4`, should also check other criteria). If `A2` appears in `Q4`, then a 3-hop reasoning chain (`Q1->Q2->Q4`) may be possible.
>
>
>
> > 3. **Generating complex questions by composing intermediate sub-questions has been explored in traditional methods [6]. Selecting demonstrations via clustering-based methods has also been investigated by existing prompting methods for other tasks [7]. This weakens the innovation of the proposed methods. I suggest the authors to shed more light on the elaborate design of the proposed method for the ODMR tasks.**
>
> Trivedi et al [6] propose to construct multi-hop questions by combining QAs from existing commonsense QA datasets. Different from their work, we propose to construct multi-step QAs with the self-generated QA pairs of LLMs, which requires neither human annotation nor existing labeled datasets. Zhang et al [7] proposed a clustering based sampling method to select CoTs for each question, which proves to be effective in our experiment settings (Table 3 in Appendix, page 12). Therefore, we adopt the same strategy to sample demonstrations for in-context learning.
>
> In this paper, we focus on elaborating the design of our self-generation framework, while only briefly introducing the reasoning chain composition (inspired by Trivedi et al [6]) in Stage 2 Step 1. Our main innovation can be summarized as follows:
>
> - An automated generation pipeline for ODMR datasets by LLMs’ self-generation, totally from scratch.
> - SP-CoT, an automated framework for mass-producing CoTs while ensuring quality and diversity, built upon the generated ODMR datasets.
> - General effectiveness of SP-CoT on larger-scale (175B) and smaller-scale (13B) LLMs and high-quality intermediate reasoning steps.
>
>
>
> > 4. **In the Method section, in stage 1, the generated questions already include answers to intermediate questions. However, in stage 2, during step 4, why are the intermediate questions replaced with corresponding answers again?**
>
> In stage 1, the generated questions indeed include answers to last-hop questions, which is to ensure the composability criteria [6]. In stage 2 step 4, the goal is to transform the reasoning chains (linked Sub-QAs) to the natural questions that require multi-step reasoning. To do this, the intermediate answers (Sub-answers) are replaced by the corresponding questions (Sub-questions) to generate the corresponding multi-step questions. Following is an example of such process:
>
> **Reasoning chain:** `Sub-Q1 -> Sub-Q3 <- Sub-Q2`
>
> - Sub-Q1: What is the American Gothic painting by Grant Wood about?
>
> - Sub-Q2: What country did Nikola Tesla emigrate to in 1884 to work for Thomas Edison in New York City?
>
> - Sub-Q3: Is it the Mississippi River that serves as the eastern border of the [*Sub-A1*] region in [*Sub-A2*]?
>
>  **Raw question:** Is it the Mississippi River that serves as the eastern border of the [*What is the American Gothic painting by Grant Wood about?*] region in [*What country did Nikola Tesla emigrate to in 1884 to work for Thomas Edison in New York City?*]?
>
> **Instruction to LLM**: Replace the sentence within [] with a relative clause and make the raw question into a natural question:
>
> **Reformulated Natural Question requiring multi-step reasoning**: Is it the Mississippi River that serves as the eastern border of the American Gothic painting by Grant Wood region in the country that Nikola Tesla emigrated to in 1884 to work for Thomas Edison in New York City?
>
>
>
> > 5. **It would be preferable if the examples presented in Figure 2 and Figure 3 in the paper could correspond to each other.**
>
> Thanks for the suggestion, the examples presented in Figure 2 and Figure 3 will be correspondant.
>
>
>
> > 6. **Recently, one of the commonly used datasets for Knowledge Base Question Answering (KBQA) is GrailQA [8], which also features multi-hop reasoning. I suggest to conduct an experiment on this dataset as well.**
>
> We report our experiment of CoT methods on 1k subset of test set provided by GrailQA [8]. According to our ODMR setting, no external knowledge is provided to LLMs. From the results below, we notice that our proposed SP-CoT is effective on GrailQA, our results on InstructGPT(`text-davinci-003`) are:
>
> | Metric | Zeroshot | Zeroshot-CoT [10] | Manual-CoT [9] (Cherry-Picked) | Manual-CoT [9] (Random) | SP-CoT (Ours) |
> | :----: | :------: | :---------------: | :----------------------------: | :---------------------: | :-----------: |
> |   EM   |   12.9   |       13.5        |              18.2              |          17.7           |     16.0      |
> |   F1   |   24.3   |       25.2        |              31.0              |          30.7           |     28.2      |
>
>
>
> [1] Trivedi, H., Balasubramanian, N., Khot, T., & Sabharwal, A. (2022). Interleaving retrieval with chain-of-thought reasoning for knowledge-intensive multi-step questions. *arXiv preprint arXiv:2212.10509*.
>
> [2] Khattab, O., Santhanam, K., Li, X. L., Hall, D., Liang, P., Potts, C., & Zaharia, M. (2022). Demonstrate-Search-Predict: Composing retrieval and language models for knowledge-intensive NLP. *arXiv preprint arXiv:2212.14024*.
>
> [3] Qu, Y., Ding, Y., Liu, J., Liu, K., Ren, R., Zhao, W. X., ... & Wang, H. (2020). RocketQA: An optimized training approach to dense passage retrieval for open-domain question answering. *arXiv preprint arXiv:2010.08191*.
>
> [4] Sachan, D. S., Patwary, M., Shoeybi, M., Kant, N., Ping, W., Hamilton, W. L., & Catanzaro, B. (2021). End-to-end training of neural retrievers for open-domain question answering. *arXiv preprint arXiv:2101.00408*.
>
> [5] Banerjee, P., & Baral, C. (2020). Knowledge fusion and semantic knowledge ranking for open domain question answering. *arXiv preprint arXiv:2004.03101*.
>
> [6] Trivedi, H., Balasubramanian, N., Khot, T., & Sabharwal, A. (2022). ♫ MuSiQue: Multihop Questions via Single-hop Question Composition. *Transactions of the Association for Computational Linguistics*, *10*, 539-554.
>
> [7] Zhang, Z., Zhang, A., Li, M., & Smola, A. (2022). Automatic chain of thought prompting in large language models. *arXiv preprint arXiv:2210.03493*.
>
> [8] Gu, Y., Kase, S., Vanni, M., Sadler, B., Liang, P., Yan, X., & Su, Y. (2021, April). Beyond IID: three levels of generalization for question answering on knowledge bases. In *Proceedings of the Web Conference 2021* (pp. 3477-3488).
>
> [9] Wei, J., Wang, X., Schuurmans, D., Bosma, M., Xia, F., Chi, E., ... & Zhou, D. (2022). Chain-of-thought prompting elicits reasoning in large language models. *Advances in Neural Information Processing Systems*, *35*, 24824-24837.
>
> [10] Kojima, T., Gu, S. S., Reid, M., Matsuo, Y., & Iwasawa, Y. (2022). Large language models are zero-shot reasoners. *Advances in neural information processing systems*, *35*, 22199-22213.

---

### Official Review · Reviewer_4LDF · 2023-08-05

**Soundness:** 3

**Excitement:**

3: Ambivalent: It has merits (e.g., it reports state-of-the-art results, the idea is nice), but there are key weaknesses (e.g., it describes incremental work), and it can significantly benefit from another round of revision. However, I won't object to accepting it if my co-reviewers champion it.

**Missing References:**

Chain-of-Thought Prompting Elicits Reasoning in Large Language Models
DEMONSTRATE–SEARCH–PREDICT: Composing retrieval and language models for knowledge-intensive NLP
Interleaving Retrieval with Chain-of-Thought Reasoning for Knowledge-Intensive Multi-Step Questions

**Paper Topic And Main Contributions:**

The author proposed an automated approach to generate the demonstration for the multi-hop reasoning QA for large language models (LLMs).

**Reasons To Accept:**

The paper provides well-designed figures that articulate each step of the methodology. These visuals enhance the clarity of the presentation, making it accessible and easy for the audience to grasp the nuances of the proposed work. This commendable effort in visualization aids in bridging the gap between complex concepts and their intuitive understanding.

**Reasons To Reject:**

The paper does not provide a comparative evaluation with the state-of-the-art (SOTA) methods in the domain of open-domain multi-hop QA. For example, ACL 2023 work Interleaving Retrieval with Chain-of-Thought Reasoning for Knowledge-Intensive Multi-Step Questions. And 2022 work DEMONSTRATE–SEARCH–PREDICT: Composing retrieval and language models for knowledge-intensive NLP. On those works, the performance on HotpotQA is around EM 49 F1 60, which is much higher than this work SP-CoT EM 33, F1 42.
In Table 2, the author only compared zero-shot and SP-CoT. I think the author should at least compare SP-CoT with CoT rather than zero-shot. CoT (Chain-of-Thought Prompting Elicits Reasoning in Large Language Models), where the author can human-label 8 demos.

**Reproducibility:**

3: Could reproduce the results with some difficulty. The settings of parameters are underspecified or subjectively determined; the training/evaluation data are not widely available.

**Reviewer Confidence:**

5: Positive that my evaluation is correct. I read the paper very carefully and I am very familiar with related work.

---

> ### Author Rebuttal · Authors · 2023-08-29
>
> Thanks for your feedback and thoughtful suggestions.
>
> > 1. **The paper does not provide a comparative evaluation with the state-of-the-art (SOTA) methods in the domain of open-domain multi-hop QA. For example, ACL 2023 work Interleaving Retrieval with Chain-of-Thought Reasoning for Knowledge-Intensive Multi-Step Questions. And 2022 work DEMONSTRATE–SEARCH–PREDICT: Composing retrieval and language models for knowledge-intensive NLP. On those works, the performance on HotpotQA is around EM 49 F1 60, which is much higher than this work SP-CoT EM 33, F1 42.**
>
> We afraid that this concern comes from misunderstanding of our work setting, actually, our research scope is subject to a challenging and quite different setting,  multi-step question-answering without quoting any external knowledge source. Different from previous knowledge-dependent works in ACL 2022 [2] and ACL 2023 [1], we explore the limit capability of LLMs for multi-step reasoning via self-enhancement. It’s worth noting that our method is “**LLM-only**” for open-domain multi-hop QA, which does not require external knowledge. Admittedly, such works [1,2] will be added in Table 1 as SOTA results of retrieval-based methods for reference, rather than comparison baselines.
>
> In the following table, we report more reference results on our proposed ODMR task, including latest knowledge-dependent methods [1,2] and our implementation of Retrieval-based methods on most recent LLMs (We use DPR to retrieve top-5 documents from Wikipedia as context and employ LLM as Reader to answer the question based on the context).  The scores are EM/F1 on the full dev set of 4 ODMR benchmarks.
>
> |                            Method                            | MusiqueQA | HotpotQA  | WikiMHQA  |   CWebQ   |   Mean    |
> | :----------------------------------------------------------: | :-------: | :-------: | :-------: | :-------: | :-------: |
> |              DPR+ChatGPT (`gpt-3.5-turbo-0301`)              |  1.7/4.1  | 15.8/21.4 | 10.9/18.1 | 13.7/18.7 | 10.5/15.6 |
> |             DPR+InstructGPT (`text-davinci-003`)             | 4.8/11.6  | 26.3/34.8 | 23.3/27.1 | 34.4/41.6 | 22.2/28.8 |
> |                       DPR+Alpaca(13B)                        |  2.2/8.4  | 12.0/21.5 | 12.6/21.2 | 15.9/27.1 | 10.7/19.5 |
> |                       DPR+Vicuna(13B)                        |  2.2/7.4  | 18.6/25.8 | 23.9/27.6 | 20.3/27.9 | 16.2/22.1 |
> |                       DPR+Wizard(13B)                        | 3.5/10.0  | 19.9/28.4 | 22.8/27.5 | 24.2/32.2 | 17.6/24.5 |
> |                      IRCoT (GPT-3) [1]                       |  -/30.8   |  -/59.1   |  -/66.5   |    -/-    |    -/-    |
> | DEMONSTRATE–SEARCH–PREDICT (Codex(`code-davinci-002`)) [2] |    -/-    | 51.4/62.9 |    -/-    |    -/-    |    -/-    |
> |       SP-CoT (Ours, ChatGPT (`gpt-3.5-turbo-0301`))       | 14.5/22.6 | 33.2/42.9 | 30.1/34.7 | 37.5/43.6 | 28.8/36.0 |
>
>
>
> > 2. **In Table 2, the author only compared zero-shot and SP-CoT. I think the author should at least compare SP-CoT with CoT rather than zero-shot. CoT (Chain-of-Thought Prompting Elicits Reasoning in Large Language Models), where the author can human-label 8 demos.**
>
> To provide a more comprehensive picture of current CoT methods on ODQA, we report hereby the performance (EM scores) of additional CoT variants, including Manual-CoT [3] and Auto-CoT [4] for all LLMs mentioned in this paper.  In our experiment, Manual-CoT (Cherry-Pick) adopts 8 cherry-picked questions and their CoTs manually writen by the authors. The results of Manual-CoT (Random) report the mean EM scores of randomly selected questions and theirs manual CoTs for 2 experiments across the 4 benchmarks.
>
> Following are additional results of more LLMs on 1k subsets of the test sets. The scores are the average EM scores on 4 ODMR benchmarks.
>
> |            Method            | ChatGPT(`gpt-3.5-turbo-0301`) | InstructGPT(`text-davinci-003`) | Alpaca-13B | Vicuna-13B | Wizard-13B |
> | :--------------------------: | :---------------------------: | :-----------------------------: | :--------: | :--------: | :--------: |
> |           Zeroshot           |             20.8              |              22.9               |    8.9     |    12.6    |    10.9    |
> | Manual-CoT [3] (Cherry-Pick) |             28.7              |              33.1               |    18.0    |    23.7    |    24.7    |
> |   Manual-CoT [3] (Random)    |             26.3              |              32.4               |    17.7    |    22.0    |    24.0    |
> |         Auto-CoT [4]         |             21.2              |              31.4               |    16.9    |    21.3    |    23.3    |
> |        SP-CoT (Ours)         |             26.1              |              33.1               |    17.4    |    21.5    |    22.5    |
>
> Although Manual-CoT outperformes Automated methods, it requires high quality human-labeled CoTs, which is not always accessible in real world applications. Since the cherry-picked CoTs  take the dataset features in to consideration, we consider their results as the **theoretical  upper limit** of automated approaches. Compared to previously automatic SOTA method (Auto-CoT), our proposed SP-CoT shows a decent performance boost in most cases.
>
>
>
>
>
> [1] Trivedi, H., Balasubramanian, N., Khot, T., & Sabharwal, A. (2022). Interleaving retrieval with chain-of-thought reasoning for knowledge-intensive multi-step questions. *arXiv preprint arXiv:2212.10509*.
>
> [2] Khattab, O., Santhanam, K., Li, X. L., Hall, D., Liang, P., Potts, C., & Zaharia, M. (2022). Demonstrate-Search-Predict: Composing retrieval and language models for knowledge-intensive NLP. *arXiv preprint arXiv:2212.14024*.
>
> [3] Wei, J., Wang, X., Schuurmans, D., Bosma, M., Xia, F., Chi, E., ... & Zhou, D. (2022). Chain-of-thought prompting elicits reasoning in large language models. *Advances in Neural Information Processing Systems*, *35*, 24824-24837.
>
> [4] Zhang, Z., Zhang, A., Li, M., & Smola, A. (2022). Automatic chain of thought prompting in large language models. *arXiv preprint arXiv:2210.03493*.

---

### Meta-Review · Area_Chair_cMjV · 2023-09-19

**Recommendation:** 3

**Metareview:**

The paper introduces a method to automatically generating questions via LLMs and augment it as demonstrations for open domain multi hop reasoning. Their pipeline includes reasoning chain composition, duplication control and other aspects.

Strengths:
The generation pipeline for ODMR datasets by LLMs’ self-generation and their method (SP-CoT) built upon the generated ODMR datasets. This yields good results on larger-scale (175B) and smaller-scale (13B) LLMs and high-quality intermediate reasoning steps.

Weaknesses:
- (Main concern) Comparison with (a) non-LLM approaches, and (b) with HotpotQA leaderboard systems, and (c) conceptual comparison with existing approaches with similar pipeline including decompositions (incl. recent open-domain QA with or without retrieval) ; would add depth to the paper.
- Missing ablation studies of the pipeline. This will also strengthen the paper.
Please include the newly reported results on additional CoT baseline empirical studies in the next version.

---

### Decision · Program_Chairs · 2023-10-07

**Decision:**

Accept-Findings

**Comment:**

The paper introduces a method to automatically generating questions via LLMs and augment it as demonstrations for open domain multi hop reasoning. Their pipeline includes reasoning chain composition, duplication control and other aspects.

Strengths:
The generation pipeline for ODMR datasets by LLMs’ self-generation and their method (SP-CoT) built upon the generated ODMR datasets. This yields good results on larger-scale (175B) and smaller-scale (13B) LLMs and high-quality intermediate reasoning steps.

Weaknesses:
- (Main concern) Comparison with (a) non-LLM approaches, and (b) with HotpotQA leaderboard systems, and (c) conceptual comparison with existing approaches with similar pipeline including decompositions (incl. recent open-domain QA with or without retrieval) ; would add depth to the paper.
- Missing ablation studies of the pipeline. This will also strengthen the paper.
Please include the newly reported results on additional CoT baseline empirical studies in the next version.